# White Isthmus Transcriptome Analysis Reveals the Mechanism of Translucent Eggshell Formation

**DOI:** 10.3390/ani14101477

**Published:** 2024-05-15

**Authors:** Ying Ma, Yuxing Luo, Wen Li, Dehe Wang, Zhonghua Ning

**Affiliations:** 1National Engineering Laboratory for Animal Breeding, Key Laboratory of Animal Genetics, Breeding and Reproduction, Ministry of Agriculture and Rural Affairs, College of Animal Science and Technology, China Agricultural University, Beijing 100193, China; maying@cau.edu.cn (Y.M.); l441826647@163.com (Y.L.); 15520739895@163.com (W.L.); 2Department of Animal Science and Technology, Hebei Agricultural University, Baoding 071001, China; theconcertevent@cau.edu.cn

**Keywords:** translucent eggs, transcriptome, gene expression, laying hens

## Abstract

**Simple Summary:**

Our study investigates the genetic basis of translucent eggshell formation by analyzing gene expression in the oviduct’s isthmus region of hens laying translucent eggs. Our findings highlight that alterations in collagen-related pathways are central to changes in the eggshell’s physical structure, leading to translucence. This research offers insights for strategies to reduce translucent egg formation.

**Abstract:**

The presence of translucent eggshells is a type of egg quality issue that impacts egg sales. While many researchers have studied them, the exact mechanisms behind their formation remain unclear. In this study, we conducted a transcriptomic differential expression analysis of the isthmus region of the oviduct in both normal egg- and translucent egg-laying hens. The analysis revealed that differentially expressed gene pathways were predominantly concentrated in the synthesis, modification, and transport of eggshell membrane proteins, particularly collagen proteins, which provide structural support. These findings suggest that variations in the physical structure of the eggshell membrane, resulting from changes in its chemical composition, are the fundamental cause of translucent eggshell formation. This research provides a theoretical reference for reducing the occurrence of translucent eggs.

## 1. Introduction

The presence of translucent eggshells is a quality issue associated with eggs which manifests through light brown, watermark-like spots on the surface of eggshells under certain conditions of time and temperature after egg production, distinguishing these spots from traditional black spots. Although the appearance of translucent eggshells does not affect the internal quality of eggs within the first three weeks of storage [1], it significantly impacts the visual aesthetics of eggs and reduces their market value, thereby exerting a substantial influence on production profitability [2].

Since the concept of “translucent eggs” was first introduced by Holst et al. in 1932 [3], numerous scholars have explored the mechanisms behind the formation of translucent eggshells, yet there is no consensus on the results. Talbot et al. [4] discovered that the porosity in the translucent areas of eggshells is higher and that translucent areas appear when moisture enters. Solomon [5], Chousalkar [6], and Bain [7] have all found that the shell of translucent eggs exhibits defects and more pronounced cracks in the mineralization process, suggesting that variations in the eggshell’s palisade layer contribute to the formation of translucent eggs. Subsequently, Liu et al. [8] found that the shell thickness of translucent eggs is significantly lower than that of normal eggs, and Nie et al. [9] reported similar findings; however, they suggested that uneven calcium ion deposition within the eggshell might be the primary cause of translucent eggshell formation from a nutritional perspective. In our previous investigation into the microstructure of eggshells and their associated characteristics, we observed that the eggshell membrane of translucent eggs was less thick and had a significantly lower maximum tensile strength (in the longitudinal direction) compared to normal eggs [10]; however, there were no significant differences found in either gas pore or bubble pore traits between translucent and opaque eggs in either line [1]. We speculated that differences in the physical and chemical structure of the eggshell membrane are the primary factors contributing to the formation of translucent eggs [10]. In 2023, Wang et al. [11] conducted genetic heritability calculations for eggshell translucence grades in three different hen breeds; they found heritability values of 0.30, 0.24, and 0.20, indicating moderate-to-low heritability, thereby providing an important basis for locating the genes regulating the trait of eggshell translucence. The main components of eggshell membranes are 80–85% protein, 3% liposomes, and 2% carbohydrates [12,13,14]. A wide range of protein families have been identified in the proteome of ESM, including avian beta defensins, collagens, cysteine-rich eggshell membrane proteins (CREMPs), enolase, heterogeneous ribonucleoproteins, histones, motor proteins, protein-coupled receptor kinase interactors, and zinc finger proteins. The carbohydrates in eggshell membranes include acidic glycosaminoglycans, the main components of which are hyaluronic acid, chondroitin sulfate [15,16], and dermatan sulfate. The neutral lipid fraction in eggshell membranes mainly consists of cholesterol, cholesteryl esters, and diglycerides; the minor components are free fatty acids, triglycerides, and monoglycerides [17].

For this study, we collected tissue samples from the isthmus region of the oviduct—where the eggshell membrane forms—from both normal egg-laying hens and hens that produced eggs with translucent eggshells. Using transcriptomics techniques, we conducted analyses of differentially expressed genes and gene pathway enrichment. These analyses successfully pinpointed gene pathways related to the synthesis, modification, and transport of eggshell membrane proteins. Our findings confirm that changes in the physical and chemical properties of the eggshell membrane are the root cause of translucent eggshell formation. Furthermore, we provide a genetic-level explanation for the mechanisms behind translucent eggshell formation. This research lays the groundwork for future efforts to develop methods aimed at reducing the occurrence of translucent eggs.

## 2. Materials and Methods

Previous research has demonstrated that changes in the physicochemical structure of the eggshell membrane are a key factor leading to the appearance of translucence in eggshells. The formation of the eggshell membrane primarily occurs in the hen’s oviduct isthmus. Therefore, in this experiment, we selected 6 hens with the most severe translucence and 6 normal hens from a group of 1200 hens raised under identical conditions, from which we collected tissue from the oviduct isthmus for transcriptome sequencing analysis. Through comparing the gene expression differences between translucent eggs and normal eggs, we aimed to identify and confirm the key differentially expressed genes responsible for the formation of eggshell translucence. This could provide new molecular insights into eggshell quality issues.

### 2.1. Animal Experiments

A total of 1200 Dwarf Layers (age: 40 weeks) were raised in the third level of H-type cages. The cage density was one hen per cage. Feed and water were provided ad libitum, with a photoperiod of 16 h:8 h (light/dark).

Eggs were collected continuously for the next three days. Subsequently, the eggs were stored in an egg repository maintained at a temperature of 15–20 °C and relative humidity (RH) of 50–60% for five days.

After being stored for 5 D, the eggs were classified into 4 score levels, and the scoring procedures were repeated 5 times. The translucent spot scoring followed the four-level system used by Wang De-He [18], which is based on spot size and density (Figure 1). Of these, we selected 6 hens that consistently laid normal eggs and 6 hens that consistently laid translucent eggs, as the normal egg group and the translucent egg group, respectively.

### 2.2. RNA Extraction, Library Preparation, and Sequencing

For three consecutive days, starting from 5:00 AM, both groups of experimental hens were monitored every 30 min to accurately record their egg-laying times. On the fourth day, four hours after the hens laid their eggs (the phase of eggshell calcification), the white isthmus segment of the oviduct was collected. The “white isthmus” (WI) corresponds to the upper part of the isthmus, where the eggshell membrane fibers are secreted. This region can be easily distinguished from the magnum due to a translucent ring, indicating a change in cell type [19].

Next, RNAs were extracted from the samples using TRIzol reagent (Cat. No. 155–96-026, Invitrogen Life Technologies, Carlsbad, CA, USA) according to the manufacturer’s instructions. The RNA quality and concentration were measured using a spectrophotometer (NanoDrop Technologies, Rockland, DE, USA). Qualified RNA samples were sent to Majorbio (Shanghai, China) for library construction. Transcriptome sequencing was performed using the Illumina HiSeq X Ten (Illumina, San Diego, CA, USA).

### 2.3. RNA-Seq Data Analysis

Firstly, the raw reads had their quality assessed and were filtered using the fastp software (0.20.1) [20]. Secondly, clean reads were used to make the index file, and alignment with the chicken (GRCg6a) reference-based genome was carried out using HISAT2 (2.2.1) [21]. Thirdly, the sequencing alignment/mapping (SAM) file was converted to a binary alignment/mapping (BAM) file and sorted using SAMtools (1.11) [22]. Next, the number of reads mapped to each gene was calculated using featureCounts [23], and then the DEGs between two groups were identified using the edgeR package v3.32.0 in R v4.0.3 based on the following criteria: false discovery rate (FDR) < 0.05 and |log2 fold change| > 1.5 [24]. The acquired DEGs were used to construct a heatmap to visualize the differentially expressed genes between the two groups. The DEGs were divided into upregulated and downregulated genes according to their FC value.

### 2.4. Statistical Analysis

GO [25] annotation analysis and KEGG [26] pathway enrichment analysis were performed on intersecting genes using the R package clusterProfiler (version 3.19) [27], with a threshold value of <0.05 for FDR considered statistically significant, entry screening criteria of adj. *p*-value  <  0.05 and q-value  <  0.05, and *p*-value correction using the Benjamini–Hochberg method (BH). Gene set enrichment analysis (GSEA) was performed to compare the translucent eggshell group with the normal eggshell group. The KEGG pathways with significant enrichment results were demonstrated on the basis of the NES (net enrichment score), gene ratio, and *p*-value. Gene sets with |NES| > 1, NOM *p* < 0.05, and FDR q < 0.25 were considered to show significant enrichment [28].

To enhance our precision in identifying and analyzing gene interactions, we implemented strict selection criteria for differentially expressed genes (DEGs), setting thresholds at a false discovery rate (FDR) < 0.05 and an absolute log2 fold change > 2. This step ensured that our focus was on genes that exhibit significant expression changes and are statistically significant. By adopting these criteria, we effectively enhanced the reliability and relevance of the predicted protein–protein interaction (PPI) network.

Following this, we conducted homology searches using the BLASTX tool against the genome of a closely related species, targeting these strictly selected DEGs. The latter species was chosen based on the availability of its protein–protein interaction data in the STRING database (http://string-db.org/, accessed on 12 January 2024), facilitating the prediction of PPIs associated with these DEGs [29]. The resultant PPI network was visualized and analyzed using the Cytoscape software (version 3.7.1, available at http://www.cytoscape.org/, accessed on 12 January 2024), a tool for complex network analysis and visualization [30]. This process involved uploading all mRNAs displaying non-additive expression patterns into the STRING tool to construct the network, thereby deepening our understanding of the interactions and functional relationships between these genes.

This study was approved by the Animal Care and Use Committee of China Agricultural University (permit number: AW30601202-1-1).

## 3. Results

### 3.1. Selection of Hens and Collection of Oviduct Tissue

Eggs from 1200 hens were collected over three consecutive days, totaling 3276 eggs. Of these, 1102 hens laid eggs daily. We performed translucence assessments on the eggs from these 1102 hens and found that the translucence grades were generally stable for each hen. Specifically, 32 hens consistently laid eggs graded at translucence level 1, and 112 hens consistently laid eggs graded at translucence level 4. Over the observation period, the laying intervals for 26 of the level 1 translucent egg-laying hens and 87 of the level 4 translucent egg-laying hens were consistently around 24.5 h. From each of these groups, 20 hens were selected for detailed observation, and dissections were carried out 4 h post-laying. The dissections revealed that nine of the level 1 translucent egg-laying hens and six of the level 4 translucent egg-laying hens had eggs present in the isthmus of the oviduct. Finally, six hens from each group which had eggs in the isthmus were selected for tissue sampling from this region for transcriptome sequencing analysis.

### 3.2. Construction of the Raw Reads, Mapping, and Batch Correction

After quality control, a total of 489,888,106 clean reads were obtained from 508,321,704 raw reads (96.37% of the raw reads). The reads were then mapped using HISAT2, and the unique mapping rate was 75.87 to 80.03% of reads with the chicken (GRCg6a) reference-based genome (Appendix A). The gene expression levels were illustrated using principal component analysis (PCA) (Figure 2).

### 3.3. Identification of DEGs between the Normal Egg Group (NEG) and Translucent Egg Group (TEG)

A total of 433 upregulated and downregulated DEGs were detected out of 23,998 genes using edgeR with FDR < 0.05 and log2FC ≥ 1.5, as shown in Figure 3. Here, we report the top five upregulated and downregulated DEGs: upregulated—PROK1 (prokineticin-1), FGF3 (fibroblast growth factor 3), ADPOQ (adiponectin), VGF (VGF nerve growth factor inducible), and IGDCC3 (immunoglobulin superfamily DCC subclass member 3); downregulated—OVGP1 (oviductal glycoprotein 1), CSAG3 (chondrosarcoma-associated gene 3), PIGR (polymeric immunoglobulin receptor), CHGA (chromogranin A), and NTS (neurotensin). The results regarding the detected genes’ expression levels, along with their FDR values, are provided in Appendix A.

### 3.4. Function and Pathway Analysis of the DEGs

The 433 identified DEGs were used to perform Gene Ontology (GO) analysis for functional annotation, and gene set enrichment analysis (GSEA) was conducted to explore their function. In the WI, 27 GO terms (10 biological processes (BPs), 7 cellular components (CCs), and 10 molecular functions (MFs)) were enriched in the extracellular region, system development, and signaling receptor binding (Figure 4a,b). In addition, the GSEA revealed the significant upregulation of the apelin signaling pathway, MAPK signaling pathway, motor proteins, and vascular smooth muscle contraction (Figure 5a), along with the significant downregulation of glycosaminoglycan biosynthesis–chondroitin sulfate/dermatan sulfate, mannose-type O-glycan biosynthesis, nucleocytoplasmic transport, protein export, protein processing in the endoplasmic reticulum, ribosomes, ribosome biogenesis in eukaryotes, and the spliceosome (Figure 5b).

### 3.5. Integration of PPI Network and Module Analysis

The DEG network interaction analysis of the white isthmus is shown in Figure 6, containing 133 genes and 103 interaction relationships. The top six genes in the white isthmus were COL3A1, COL1A1, COL5A1, OGN, ASPN, and FBN1. These genes may play an important regulatory role in the formation of eggshells.

## 4. Discussion

Our previous studies speculated that the main reason for the formation of translucence was the variation in the physical structure caused by differences in the chemical composition of the eggshell membrane. We found that the eggshell membrane of translucent eggs is thinner and more easily broken than that of normal eggs [1,31]. As a result, vapors in the contents can more easily penetrate these eggshells. Furthermore, when factors such as temperature differences lead to changes in the volume of the egg contents, the eggshell membrane weakens, becomes porous, or even ruptures. The contents of the egg then permeate through the eggshell membrane and continue to deposit between the eggshell’s microstructures through vertical diffusion via pores. When the force of outward liquid diffusion and the moisture evaporation on the eggshell surface reach equilibrium, scattered translucent spots of varying size appear on the eggshell surface.

In this study, we performed transcriptome analysis to explore the DEGs in the ovaries of hens that laid translucent eggs and hens that laid normal eggs. We prepared a cDNA library and performed RNA sequencing for each blood sample; during the analysis, we detected 433 mRNA transcripts: 380 upregulated and 53 downregulated. Differential expression analysis identified the top five upregulated and downregulated genes, which are as follows: upregulated—PROK1 (prokineticin-1), FGF3 (fibroblast growth factor 3), ADIPOQ (adiponectin), VGF (VGF nerve growth factor inducible), and IGDCC3 (immunoglobulin superfamily DCC subclass member 3); downregulated—OVGP1 (oviductal glycoprotein 1), PIGR (polymeric immunoglobulin receptor), CHGA (chromogranin A), NTS (neurotensin), and CSAG3 (chondrosarcoma-associated gene 3).

A previous study showed that PROK1 acting via PROKR1 increased the proliferation and adhesion trophoblast cells in a manner mediated by the MAPK and/or PI3K/AKT signaling pathways to promote follicle maturation and development [32]. FGF3 plays an important role in the regulation of embryonic development, cell proliferation, and cell differentiation. ADIPOQ (adiponectin) is expressed exclusively in adipose tissue; it encodes a protein similar to collagens X and VIII and complement factor C1q [33]. Collagen is a structural protein that serves as a crucial structural component in various tissues [34], including structures such as eggshell membranes. VGF (VGF nerve growth factor inducible) is specifically expressed in a subpopulation of neuroendocrine cells and is upregulated by nerve growth factor [35,36]. IGDCC3 is a protein-coding gene that is predicted to be an integral component of the plasma membrane [37].

OVGP1 is expressed in the endometrial epithelium and plays a crucial role in regulating receptivity and aiding trophoblast adhesion; it is specifically induced in the luminal epithelium at the time of embryo implantation, where it regulates genes associated with receptivity and assists in the adhesion of the trophoblast cell line in vitro [38]. The PIGR gene is responsible for the transport of polymeric IgA across various mucosal epithelial layers [39]. Chromogranin A is known to be a protein encoded by the CHGA gene in humans; it is a member of the granin family of neuroendocrine secretory proteins and is found in the secretory vesicles of neurons and endocrine cells [40].

Pathway enrichment analysis showed that the “Extracellular matrix organization” pathway was highly significant, and the proportion of enriched genes exceeded 0.4. The extracellular matrix is a component of animal tissues, primarily consisting of fibrous proteins such as collagen, elastin, associated microfibrils, fibronectin, and laminins embedded in a viscoelastic gel of anionic proteoglycan polymers; it plays a crucial structural role [41]. In the study conducted by Scott and Haigh in 1985, it was observed that collagen, renowned for its tensile strength, confers tissues with the capacity to recover after undergoing stretching [42]. Furthermore, structural components closely related to collagen fibers include chondroitin sulfate, dermatan sulfate, and keratan sulfate proteoglycans. The functionalities and constituents of the extracellular matrix exhibit a high degree of similarity to those of the eggshell membrane protein, which plays a pivotal role in the biological functions of the eggshell. Previous investigations have revealed alterations in the structural composition of the eggshell membrane in translucent eggs, leading to a diminished maximum longitudinal tensile strength in comparison to normal eggs [10]. This underscores the significance of the structural constituents of the eggshell membrane in influencing the formation of translucence in eggs, with the “Extracellular matrix organization” pathway deemed to play a vital role in this process. Similarly, pathways significantly enriched in tissue development, system development, anatomical structure formation involved in morphogenesis, the positive regulation of cell communication, and other pathways all involve the development of complex systems and tissues within organisms, playing crucial roles in various biological functions and the formation of eggshell membranes.

“External encapsulating structure organization” was another significantly enriched pathway, and its function is defined as the cellular process for the assembly, arrangement, or disassembly of external structures surrounding the entire cell. One of its child terms is “egg chorion assembly”, which has been reported to be involved in the formation of the eggshell membrane [43]. The biological process pathway “collagen fibril organization” and the molecular function pathway “Collagen binding” are both involved in the arrangement, binding, and interactions of collagen proteins. As mentioned above, collagen is a structural protein that is predominantly present in animal tissues and plays a crucial role in maintaining the structural strength and elasticity of tissues. These significantly enriched pathways further emphasize the essential role of the eggshell membrane structure in the production of translucent eggs.

Gene set enrichment analysis (GSEA) is a bioinformatics method employed for the interpretation of gene expression data. In contrast to traditional enrichment analysis, GSEA prioritizes the collective expression patterns of entire gene sets over the individual significance of single genes [28]. According to the GSEA results, biosynthesis and metabolic pathways were enriched. Interestingly, the pathways that were significantly enriched may be related to the modification, synthesis, and transport of proteins.

For example, four of the downregulated pathways—the spliceosome, ribosomes, ribosome biogenesis in eukaryotes, and protein processing in endoplasmic reticulum—were related to protein synthesis. Specifically, after transcription, mRNA precursors contain introns and exons. Non-coding intronic sequences are excised, and exons are joined by a macromolecular complex—the spliceosome [44]. Following this, ribosomes recognize the codons on mRNA and, with the assistance of the appropriate tRNA molecules, synthesize proteins [45]. Then, proteins are folded with the assistance of luminal chaperones. Newly synthesized peptides enter the endoplasmic reticulum (ER) through the Sec61 pore and undergo glycosylation [46].

Two other downregulated pathways (glycosaminoglycan biosynthesis–chondroitin sulfate/dermatan sulfate and mannose-type O-glycan biosynthesis) both involve protein modification. In the former pathway, chondroitin sulfate (CS), a natural anionic mucopolysaccharide, acts as the primary element of the extracellular matrix (ECM) of diverse organisms [47]; it also modifies collagen in the developing ovarian follicles of birds, forming chondroitin sulfate-modified collagen, also known as ggBM1 (Gallus gallus basement membrane protein 1). This protein is a major component between the vascularized follicular membrane and epithelioid granulosa cells [48]; it eventually forms type IV collagen networks as the follicles develop, which are important components of the eggshell membrane [49]. These type IV collagen networks have high tensile strength and provide mechanical support to tissues, including the eggshell membrane [50]. Dermatan sulfate (DS), another type of glycosaminoglycan, is also involved. Research indicates that the connection between tenascin-X and collagen is mediated by the DS of decorin [51,52], and this bridging action is crucial for establishing the normal tensile strength of the skin [53]. The mannose-type O-glycan biosynthesis pathway involves the synthesis of mannose-type O-glycans [54], which are sugar moieties attached to proteins that participate in protein modification and cell signaling [55].

Two other pathways with a decreasing trend—nucleocytoplasmic transport and protein export—are related to protein transport. Nucleocytoplasmic transport is a critical pathway for the bidirectional transport of substances between the nucleus and the cytoplasm, including the transport of proteins, RNA, and ribonucleoproteins across the nuclear membrane [56]. Protein export is an essential pathway in cellular biology that involves the active transport of proteins from the cytoplasm to the exterior of the cell or to the periplasmic compartment in Gram-negative bacteria [57]. This process ensures the proper distribution of proteins both inside and outside the cell, thereby preserving the normal functioning of the cell.

Protein network interaction analyses identified several genes associated with eggshell formation, including COL3A1, COL1A1, COL5A1, OGN, ASPN, and FBN1. COL1A1 (type I collagen) encodes the pro-alpha1 chains of type I collagen, whose triple helix comprises two alpha1 chains and one alpha2 chain [58]. Type I collagen is a major structural component of skin, bone, tendons, and other connective tissues. COL3A1 (type III collagen) produces type III collagen, found in extensible connective tissues such as the skin, lungs, and vascular system [59]. COL5A1 (type V collagen) encodes a component of type V collagen, which is found in tissues containing type I collagen and plays a significant role in regulating the assembly of heterotypic fibers composed of both type I and type V collagen. This is crucial in tissues that require high tensile strength [60]. OGN (osteoglycin) is involved in the regulation of bone density and endothelial cell migration [61]. ASPN (asporin) is involved in the regulation of collagen fibrillogenesis and may act as a critical modulator in the pathway controlling cartilage growth and differentiation [62]. FBN1 (fibrillin-1) is a protein that plays a crucial role in the formation of elastic fibers found in connective tissue [63]. These key proteins are primarily involved in the formation of collagen fibers. ASPN plays a role in regulating the production of collagen fibers, while COL1A1 encodes the pro-α1 chain of type I collagen, COL3A1 produces type III collagen, and COL5A1 encodes a component of type V collagen. These collagens are crucial in tissues that require high tensile strength.

The eggshell membrane is a biological polymer network [64] composed of fibers randomly distributed in various directions and stacked layer by layer [65]. Collagen fibers, composed of multiple collagen molecules that are interlaced and tightly coiled, assemble into larger structural units that provide tissues with high tensile strength and toughness, maintaining and enhancing the structural integrity and functionality of tissues. Our experiments revealed that the maximum longitudinal tensile strength of the eggshell membranes from the group with translucent eggshells (0.64 ± 0.20) was significantly lower than that of the normal group (0.72 ± 0.24). Moreover, the contents of proline (7.32 ± 0.15%) and lysine (3.02 ± 0.02%) in the translucent eggshell group were significantly lower than those in the normal group (proline 7.60 ± 0.09%, lysine 3.16 ± 0.08%) (*p* < 0.05) [10]. Notably, proline, which constitutes 18.9% of the amino acids in collagen, is the most abundant amino acid. These results suggest that the collagen content in the eggshell membranes of normal eggs may be significantly higher than that in the translucent eggshell group. This is consistent with the significant differences in the expression levels of genes related to collagen formation observed in this experiment.

In summary, based on the results of our experiments and previous research, we hypothesize that the inhibited expression of genes related to collagen formation leads to decreased stability and increased susceptibility to breakage in the eggshell membranes of translucent eggs. This causes the contents of the eggs to permeate through the eggshell membrane, diffusing through the pores of the vertical crystalline layer of the shell and depositing between the microscopic structures of the eggshell. When the force of the liquid diffusing outward balances with that of the evaporation of moisture from the eggshell surface, a scattered, uneven, translucent appearance emerges on the eggshell surface.

Therefore, we hypothesize that the differential expression of these genes leads to changes in the physical structure of the eggshell membrane, subsequently causing the translucent eggshell phenomenon.

## 5. Conclusions

Transcriptome sequencing analysis revealed that decreases in the expression levels of gene pathways related to the synthesis, modification, and transport of eggshell membrane proteins are the primary cause of translucent eggs. This result confirms that the main reason for the formation of translucent eggs is the difference in the chemical composition of the eggshell membrane, which leads to changes in its physical structure, as explained at the gene pathway level.

In summary, the conducted experiment provided further insights into the mechanisms underlying the formation of translucent eggs and shed light on the underlying causes at the gene pathway level.

## Figures and Tables

**Figure 1 animals-14-01477-f001:**
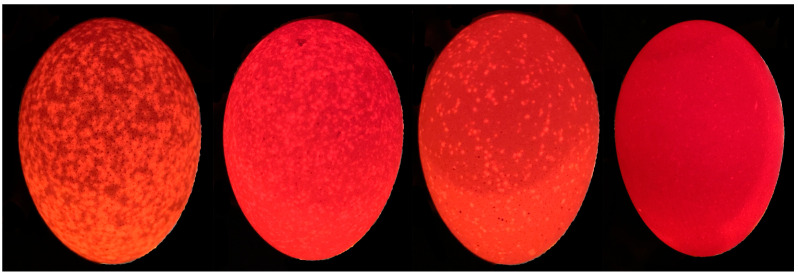
Four reference samples used for the scoring method: from left to right, scores of 4, 3, 2, and 1. Level 1: Eggs without any dark spots that showed uniform transparency when illuminated by the light source. Level 2: Eggs with small-diameter dark spots or sporadic larger dark spots that appeared as small white translucent spots when illuminated. Level 3: Eggs with larger-diameter dark spots or sporadic dark spots connected in small patches. Level 4: Eggs with numerous larger-diameter dark spots forming continuous patches covering the entire eggshell.

**Figure 2 animals-14-01477-f002:**
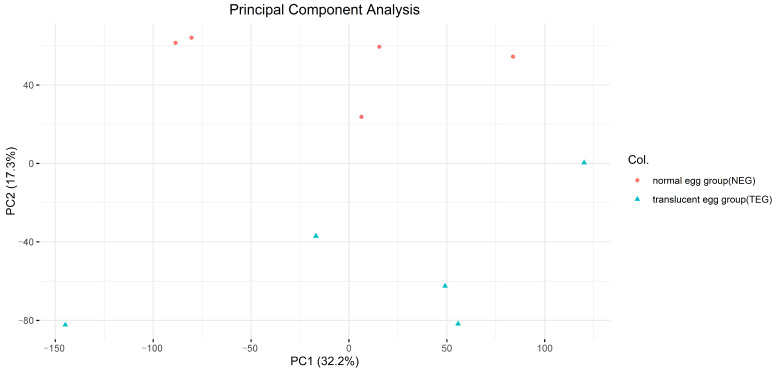
A PCA plot of the genes expressed in the white isthmus tissue samples from the normal egg group and the translucent egg group; the ellipses and shapes show the clustering of the samples.

**Figure 3 animals-14-01477-f003:**
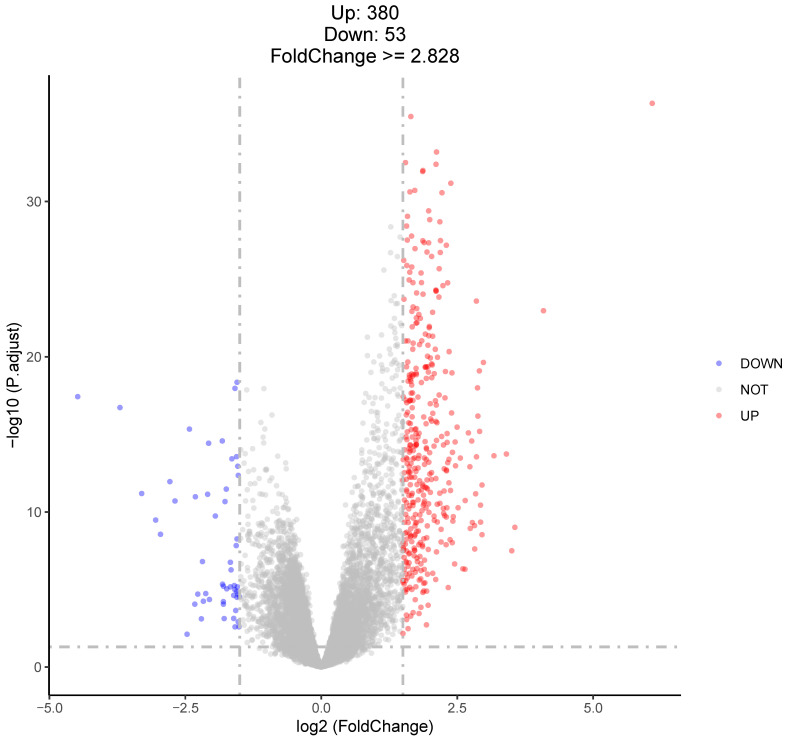
The number of DEGs in the spleen (FDR < 0.05 and log2 fold change ≥ 1.5).

**Figure 4 animals-14-01477-f004:**
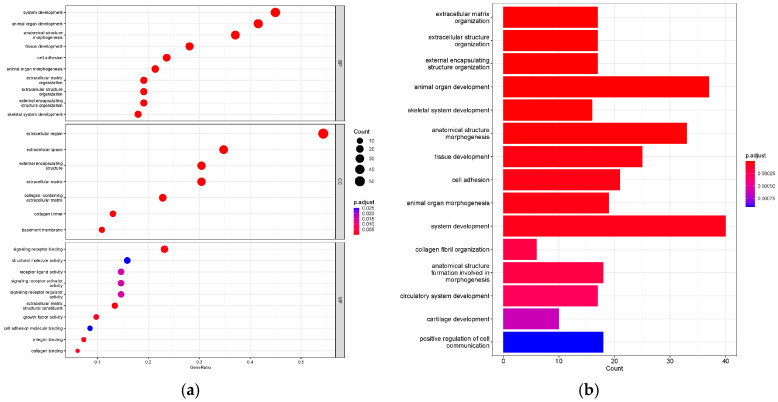
Functional enrichment analyses. GO terms (**a**) and bubble diagram of DEGs in the WI (**b**).

**Figure 5 animals-14-01477-f005:**
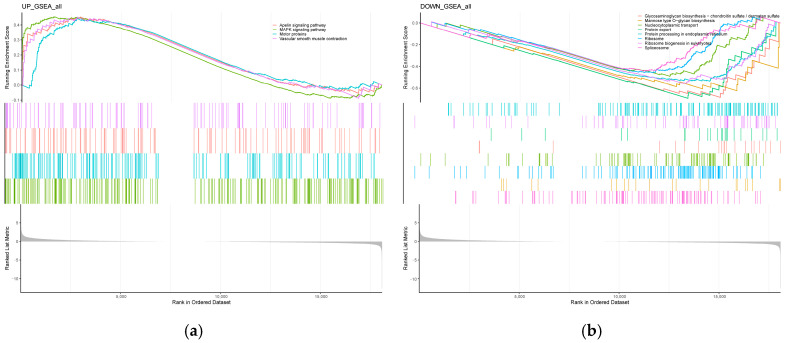
GSEA for samples from the translucent and normal egg groups. (**a**) Running enrichment scores for upregulated gene sets in the translucent egg group. Colored lines represent different pathways. The middle panel shows gene positions within each set, and the bottom panel shows the ranked gene list. (**b**) Running enrichment scores for downregulated gene sets in the translucent egg group. Colored lines represent different pathways. The middle panel shows gene positions within each set, and the bottom panel shows the ranked gene list.

**Figure 6 animals-14-01477-f006:**
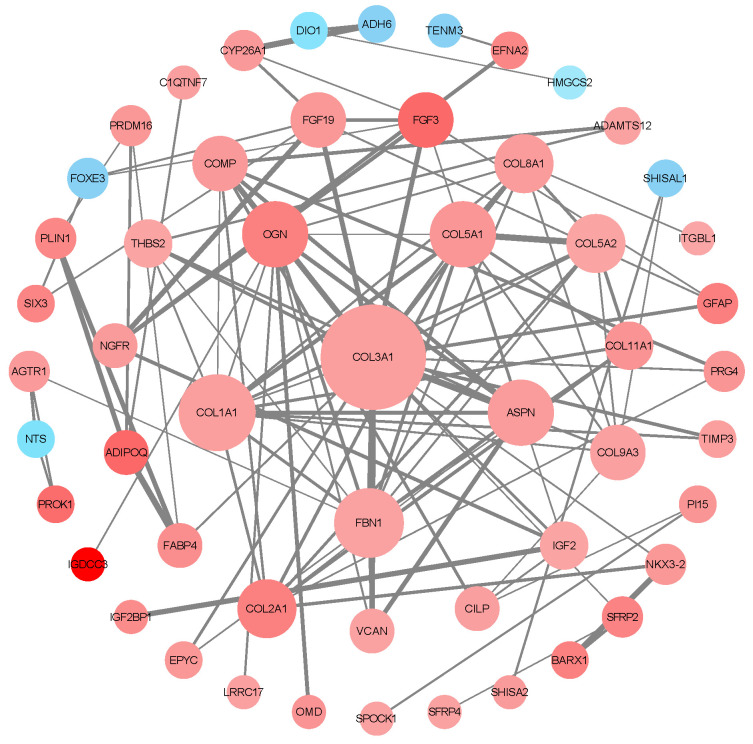
Protein-protein interaction (PPI) network for DEGs in white isthmus. This PPI network map de-picts the interactions among proteins, where nodes represent proteins and edges indicate interac-tions. Node colors signal regulatory changes—blue for downregulation, red for upregulation, with color intensity reflecting the magnitude of regulation. Node size corresponds to the number of interactions, with larger nodes representing highly connected protein ‘hubs’.

## Data Availability

The original contributions presented in the study are included in the article/Appendix A, further inquiries can be directed to the corresponding author.

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
