# Peer review of "White Isthmus Transcriptome Analysis Reveals the Mechanism of Translucent Eggshell Formation"

_animals, 2024, doi:10.3390/ani14101477_

Round 1
Reviewer 1 Report
Comments and Suggestions for Authors
1. The Simple Summary is too long and tedious; it is suggested to rewrite to briefly describe aims, major findings, and conclusions instead of looking like introduction.
2. Some grammars and writings should be revised and polished. Here are some examples in the abstract and title.
Ex. line 23-24, Despite many studies have been tried, the exact mechanisms of behind translucent eggshell formation remain unclear.
Ex. line 25-26, using the oviductal isthmus from normal or translucent egg-laying hens.
Ex. line 26-29, The results showed that the differentially expressed genes were predominantly annotated in the pathways involved in the synthesis, modification…..
Ex. line 31, The present results provide…..
3. DEG network interaction analysis, at least one of the top 6 gens, COL3A1, COL1A1,COL5A1, OGN, ASPN, or FBN1 should be validated in protein level by immunoblot, ELISA or immunohistochemistry.
4. Also, the content of COL3A1, COL1A1,COL5A1, OGN, ASPN, or FBN1, or total collagen content should be assessed to manifest the difference of normal and translucent eggshell eggs.
Comments on the Quality of English LanguageModerate editing of English language required
Author Response
For research article
|
Response to Reviewer 1 Comments
|
||
|
1. Summary |
|
|
|
Thank you very much for taking the time to review this manuscript. Please find the detailed responses below and the corresponding revisions/corrections highlighted/in track changes in the re-submitted files.
|
||
|
2. Questions for General Evaluation |
Reviewer’s Evaluation |
Response and Revisions |
|
Does the introduction provide sufficient background and include all relevant references? |
Yes/Can be improved/Must be improved/Not applicable |
corresponding response in the point-by-point |
|
Are all the cited references relevant to the research? |
Yes/Can be improved/Must be improved/Not applicable |
corresponding response in the point-by-point |
|
Is the research design appropriate? |
Yes/Can be improved/Must be improved/Not applicable |
corresponding response in the point-by-point |
|
Are the methods adequately described? |
Yes/Can be improved/Must be improved/Not applicable |
corresponding response in the point-by-point |
|
Are the results clearly presented? |
Yes/Can be improved/Must be improved/Not applicable |
corresponding response in the point-by-point |
|
Are the conclusions supported by the results? |
Yes/Can be improved/Must be improved/Not applicable |
corresponding response in the point-by-point |
|
3. Point-by-point response to Comments and Suggestions for Authors
|
||
|
Comments 1: The Simple Summary is too long and tedious; it is suggested to rewrite to briefly describe aims, major findings, and conclusions instead of looking like introduction. |
||
|
Response 1: Thank you for pointing this out. We agree with this comment and have changed it like this: ” Our study investigates the genetic basis of translucent eggshell formation by analyzing gene expression in the oviduct's isthmus region of hens laying translucent eggs. Our findings highlight that alterations in collagen-related pathways are central to changes in the eggshell's physical structure, leading to translucence. This research offers insights for strategies to reduce translucent egg formation. |
||
|
Comments 2: Some grammars and writings should be revised and polished. Here are some examples in the abstract and title. |
||
|
Response 2: Thank you for pointing this out. We agree with this comment. We have already undergone paid English editing and proofreading services, correcting any errors in the document. Comments 3: DEG network interaction analysis, at least one of the top 6 gens, COL3A1, COL1A1,COL5A1, OGN, ASPN, or FBN1 should be validated in protein level by immunoblot, ELISA or immunohistochemistry; Also, the content of COL3A1, COL1A1,COL5A1, OGN, ASPN, or FBN1, or total collagen content should be assessed to manifest the difference of normal and translucent eggshell eggs. Response 3: Thank you for pointing this out. We acknowledge the importance of validating the protein levels of the top genes such as COL3A1, COL1A1, COL5A1, OGN, ASPN, and FBN1 through methods like immunoblot, ELISA, or immunohistochemistry to strengthen our findings. Unfortunately, due to the depletion of our experimental materials and the time constraints associated with my impending graduation, we are currently unable to perform these additional experiments. We understand the significance of these validations and are considering alternative approaches to address this limitation. These include collaborating with another lab that could perform the required assays or planning future studies to validate these findings post-graduation We appreciate your understanding and would be grateful for any suggestions on how we might proceed under these circumstances. We are committed to ensuring the integrity and scientific validity of our study and hope to find a feasible solution.
|
||
|
4. Response to Comments on the Quality of English Language We have already undergone paid English editing and proofreading services, correcting any errors in the document. |
||
Reviewer 2 Report
Comments and Suggestions for Authors
This could be an important addition to the literature. However, there are major issues which need to be addressed.
1. Line 78-154 Materials and methods. It is not clear what is the experimental design. This makes reviewing this paper impossible.
2. Results section, it is not clear what are being compared. This makes reviewing this paper impossible.
3. Were the statistical tools employed for assessing presumptive differences between oviductal transcripts and degrees of egg shell transparency?
· If not, why not?
· If yes, please show data in a simple table in the results section.
· What was the replication?
Other issues
The authors need to thoroughly review their manuscript for English usage and what a good manuscript contains.
1. Line 110 Please do not start sentences with “And”.
2. Lines 166-176. I’d like to Supplementary Table S2 or the key information be included in the manuscript.
3. There are two figure 2s.
4. Figure 2. Please increase font size of key and include the gene names for the transcripts.
5. Figure 3, 4 and 5 Font size are unreadable.
6. Figure legends need to be expanded such that each is understandable without reference to the narrative of the text.
7. Did the hens consistently lay eggs with the same degree of transparency (using the scoring system in Figure 1).
Comments on the Quality of English LanguageSee comments to authors.
Author Response
For research article
|
Response to Reviewer 2 Comments
|
||
|
1. Summary |
|
|
|
Thank you very much for taking the time to review this manuscript. Please find the detailed responses below and the corresponding revisions/corrections highlighted/in track changes in the re-submitted files.
|
||
|
2. Questions for General Evaluation |
Reviewer’s Evaluation |
Response and Revisions |
|
Does the introduction provide sufficient background and include all relevant references? |
Yes/Can be improved/Must be improved/Not applicable |
corresponding response in the point-by-point |
|
Are all the cited references relevant to the research? |
Yes/Can be improved/Must be improved/Not applicable |
corresponding response in the point-by-point |
|
Is the research design appropriate? |
Yes/Can be improved/Must be improved/Not applicable |
corresponding response in the point-by-point |
|
Are the methods adequately described? |
Yes/Can be improved/Must be improved/Not applicable |
corresponding response in the point-by-point |
|
Are the results clearly presented? |
Yes/Can be improved/Must be improved/Not applicable |
corresponding response in the point-by-point |
|
Are the conclusions supported by the results? |
Yes/Can be improved/Must be improved/Not applicable |
corresponding response in the point-by-point |
|
3. Point-by-point response to Comments and Suggestions for Authors
|
||
|
Comments 1: Line 78-154 Materials and methods. It is not clear what is the experimental design. This makes reviewing this paper impossible. |
||
|
Response 1: Thank you for pointing this out. We agree with this comment. Therefore, we have added a description of the experimental design in lines 82-91. “Previous research has demonstrated that changes in the physicochemical structure of the eggshell membrane are a key factor leading to the appearance of translucence in eggshells. The formation of the eggshell membrane primarily occurs in the hen's ovi-duct isthmus. Therefore, in this experiment, we selected 6 hens with the most severe translucence and 6 normal hens from a group of 1200 hens raised under identical condi-tions, from which we collected tissue from the oviduct isthmus for transcriptome se-quencing analysis. Through comparing the gene expression differences between trans-lucent eggs and normal eggs, we aimed to identify and confirm the key differentially expressed genes responsible for the formation of eggshell translucence. This could pro-vide new molecular insights into eggshell quality issues.” |
||
|
Comments 2: Results section, it is not clear what are being compared. This makes reviewing this paper impossible. |
||
|
Response 2: Thank you for pointing this out. We agree with this comment. Therefore, we have added a description of the experimental design in lines 168-180. “3.1. Selection of Hens and Collection of Oviduct Tissue Eggs from 1200 hens were collected over three consecutive days, totaling 3276 eggs. Of these, 1102 hens laid eggs daily. We performed translucence assessments on the eggs from these 1102 hens and found that the translucence grades were generally stable for each hen. Specifically, 32 hens consistently laid eggs graded at translucence level 1, and 112 hens consistently laid eggs graded at translucence level 4. Over the observation pe-riod, the laying intervals for 26 of the level 1 translucent-egg hens and 87 of the level 4 translucent-egg hens were consistently around 24.5 hours. From each of these groups, 20 hens were selected for detailed observation, and dissections were carried out 4 hours post-laying. The dissections revealed that nine of the level 1 translucent-egg hens and six of the level 4 translucent-egg hens had eggs present in the isthmus of the oviduct. Finally, six hens from each group, which had eggs in the isthmus, were selected for tis-sue sampling from this region for transcriptome sequencing analysis.” Comments 3: Were the statistical tools employed for assessing presumptive differences between oviductal transcripts and degrees of egg shell transparency? · If not, why not? · If yes, please show data in a simple table in the results section. · What was the replication? Response 3: Thank you for pointing this out. In the "Materials and Methods" section, we provided detailed descriptions of the specific statistical methods employed for each result. For instance, in selecting differentially expressed genes, we utilized FDR (False Discovery Rate) statistical correction, with a filtering criterion of false discovery rate (FDR) < 0.05 and |log2 Fold Change| > 1.5, as the threshold for significant differential gene expression. Comments 4: Other issues Response 4: Thank you for pointing this out. Issues 1-6 have been corrected. As for issue 7, The same chicken will consistently produce eggs with the same level of translucence.(Described in response 1) |
||
|
4. Response to Comments on the Quality of English Language We have already undergone paid English editing and proofreading services, correcting any errors in the document. |
||

Reviewer 3 Report
Comments and Suggestions for Authors
Ovarian transcriptome analysis reveals the mechanism of translucent eggshell formation
Dear Authors,
The manuscript is very interesting and describes important issue which is mechanism of translucent eggshell formation, which can be also related to environmental conditions in the months of high air temperature and heat stress of laying hens, what affects the quality of the egg shell and its parameters during transport.
Text is readable and there are not many aspects to correct. Important element is composition of diet used during 40th week of life of layer hens. For sure important aspect in future is to taking into a consider to increase number of observations in each treatment/experimental group, minimally 10, optimally 38 with independent design for two treatments in T-test convention.
Below I add some suggestions helpful in this process:
Line 40-330
Space required before reference, ie.: storage [1].
Line 35-76
Introduction section can be expand. Maybe one paragraph about heat stress and absorption of nutrients (protein, energy, Ca, P) can be added what can have effect on lower plasma lever for sure Ca in case of birds.
Line 44.5
In footer, left side, year and volume must be changed: 2024, 14,….
Line 79
Correction required. In text is w1as, instead of was.
Line 82
Information about month and environmental temperature is will be valuable information, because in Materials and Methods is given range of temperature from 20-25°C, when for hens 20°C is maximal.
In this case heat stress could induce wase quality of eggs. Climatization cannot decrease level of heat inside and that also could have effect for absorption of nutrients (lower level in blood plasma). In this section information about diet will be valuable information (ingredient composition and chemical composition).
Line 85 and 86
In sentences is: ‘Eggs will be..’ (future form), must be (in past form): ‘…eggs were collected/stored…’
Line 88
Double dots.
Line 92
This part is strange for me. Because in experiment is available 1200 hens in 40 week of life and in final result only 6 hens from control group and 6 from translucent group are randomly assigned to final analysis. Maybe better in next experiment to increase number of animals in each replication from 1 to 4-6 animals and fit in higher level genetic material to whole “population”, because in case of 6 single observations in treatment individual variability can have influence for results for whole flock in one rear cycle of laying hens.
Power of a test also will be very low in case of 6 observations/repetitions in each treatment.
In text is chickens, but there hens must be emphasized, 40 weeks it is still very low age when possible age of hens in nature is taking into a consider, but they are not as young as broiler chickens.
Line 129
Subsection title.
In text of manuscript is: ‘Bioinformatics analysis’, but Statistical analysis is better title. In text of this section different statistics are used to read/observe final value of probability to verifying null hypothesis.
Line 137
Maybe p-value as value from sample can be better description than P-value for whole set of data from flock of layer hens in 40th week of life?
Line 169-175
Spaces between DEG’s abbreviations add explanations in brackets are required.
Line 374-507
Abbreviation of Journal’ s name is required.
Dots on the end of abbreviation required (ie. Poult. Sci.)
Line 389-392
References are specified in Chinese language. According to Instructions for authors, they can be translated to English language (or if English translation is already stated can be also added), and on the end before doi number in brackets information can be added: (in Chinese).
Author Response
For research article
|
Response to Reviewer 3 Comments
|
||
|
1. Summary |
|
|
|
Thank you very much for taking the time to review this manuscript. Please find the detailed responses below and the corresponding revisions/corrections highlighted/in track changes in the re-submitted files.
|
||
|
2. Questions for General Evaluation |
Reviewer’s Evaluation |
Response and Revisions |
|
Does the introduction provide sufficient background and include all relevant references? |
Yes/Can be improved/Must be improved/Not applicable |
corresponding response in the point-by-point |
|
Are all the cited references relevant to the research? |
Yes/Can be improved/Must be improved/Not applicable |
corresponding response in the point-by-point |
|
Is the research design appropriate? |
Yes/Can be improved/Must be improved/Not applicable |
corresponding response in the point-by-point |
|
Are the methods adequately described? |
Yes/Can be improved/Must be improved/Not applicable |
corresponding response in the point-by-point |
|
Are the results clearly presented? |
Yes/Can be improved/Must be improved/Not applicable |
corresponding response in the point-by-point |
|
Are the conclusions supported by the results? |
Yes/Can be improved/Must be improved/Not applicable |
corresponding response in the point-by-point |
|
3. Point-by-point response to Comments and Suggestions for Authors
|
||
|
Comments 1: Grammatical and formatting errors in Line 40-330, Line 44.5,Line 79, Line 85 and 86, Line 88, Line 129, Line 169-175, Line 374-507, Line 389-392 |
||
|
Response 1: Thank you for pointing this out. We agree with this comment and has corrected all of them. |
||
|
Comments 2: Information about month and environmental temperature is will be valuable information, because in Materials and Methods is given range of temperature from 20-25°C, when for hens 20°C is maximal. |
||
|
Response 2: Thank you for pointing this out. We agree with this comment. Upon verifying the original information, it was found that the storage temperature for eggs should be 15-20°C. This has been corrected. Comments 3: This part is strange for me. Because in experiment is available 1200 hens in 40 week of life and in final result only 6 hens from control group and 6 from translucent group are randomly assigned to final analysis. Maybe better in next experiment to increase number of animals in each replication from 1 to 4-6 animals and fit in higher level genetic material to whole “population”, because in case of 6 single observations in treatment individual variability can have influence for results for whole flock in one rear cycle of laying hens. Response 3: Thank you for pointing this out. We have added specific details about the process and results of sample selection in the Results section, explaining why only 6 chickens were ultimately chosen for each group. “3.1. Selection of Hens and Collection of Oviduct Tissue Eggs from 1200 hens were collected over three consecutive days, totaling 3276 eggs. Of these, 1102 hens laid eggs daily. We performed translucence assessments on the eggs from these 1102 hens and found that the translucence grades were generally stable for each hen. Specifically, 32 hens consistently laid eggs graded at translucence level 1, and 112 hens consistently laid eggs graded at translucence level 4. Over the observation period, the laying intervals for 26 of the level 1 translucent-egg hens and 87 of the level 4 translucent-egg hens were consistently around 24.5 hours. From each of these groups, 20 hens were selected for detailed observation, and dissections were carried out 4 hours post-laying. The dissections revealed that nine of the level 1 translucent-egg hens and six of the level 4 translucent-egg hens had eggs present in the isthmus of the oviduct. Finally, six hens from each group, which had eggs in the isthmus, were selected for tissue sampling from this region for transcriptome sequencing analysis.” Comments 4: Maybe p-value as value from sample can be better description than P-value for whole set of data from flock of layer hens in 40th week of life? Response 4: Thank you for pointing this out. In GSEA, the p-value calculation only involves data from 12 samples and does not involve data from the entire flock of layer hens in their 40th week of life. |
||
|
4. Response to Comments on the Quality of English Language We have already undergone paid English editing and proofreading services, correcting any errors in the document. |
||

Round 2
Reviewer 1 Report
Comments and Suggestions for Authors
If the authors ran out of the uterius samples, the the content of COL3A1, COL1A1,COL5A1, OGN, ASPN, or FBN1, or total collagen content in eggshell membrance between normal and translucent eggshell eggs sholud be assessed to confirm the results.
Comments on the Quality of English LanguageMinor editing of English language required
Author Response
Dear Reviewer,
First, we would like to express our gratitude for your thorough review and valuable comments on our research. We fully acknowledge the importance of validating our findings, and understand your suggestion to test the levels of COL3A1, COL1A1, COL5A1, OGN, ASPN, or FBN1 proteins, as well as the total collagen content in eggshell membranes.
However, we have previously attempted to quantify these specific proteins in eggshell membranes. Due to solubility issues with the membrane, our attempts did not yield satisfactory results, preventing accurate measurements of many proteins. Therefore, we were unable to use this method to validate our research findings. Our team has analyzed the microstructure of eggshells and the amino acid content of the membranes, which has allowed us to verify our results to some extent. We have included these discussions in the discussion section of our paper:
The eggshell membrane is a biological polymer network [31] composed of fibers randomly distributed in various directions and stacked layer by layer [19]. Collagen fibers, composed of multiple collagen molecules interlaced and tightly coiled, assemble into larger structural units that provide tissues with high tensile strength and toughness, maintaining and enhancing the structural integrity and functionality of tissues. Our experiments revealed that the maximum longitudinal tensile strength of the eggshell membranes from the group with translucent eggshells (0.64±0.20) was significantly lower than that of the normal group (0.72±0.24). Moreover, the contents of proline (7.32±0.15%) and lysine (3.02±0.02%) in the translucent eggshell group were significantly lower than those in the normal group (proline 7.60±0.09%, lysine 3.16±0.08%) (p<0.05). Notably, proline, which constitutes 18.9% of the amino acids in collagen, is the most abundant amino acid. These results suggest that the collagen content in the eggshell membranes of normal eggs may be significantly higher than that in the translucent eggshell group. This is consistent with the significant differences in the expression levels of genes related to collagen formation observed in this experiment.
In summary, based on the results of our experiments and previous research, we hypothesize that the inhibited expression of genes related to collagen formation leads to decreased stability and increased susceptibility to breakage in the eggshell membranes of translucent eggs. This causes the contents of the eggs to permeate through the eggshell membrane, diffusing through the pores of the vertical crystalline layer of the shell and depositing between the microscopic structures of the eggshell. When the force of the liquid diffusing outward balances with that of the evaporation of moisture from the eggshell surface, a scattered, uneven, translucent appearance emerges on the eggshell surface.
Therefore, we hypothesize that differential expression of these genes leads to changes in the physical structure of the eggshell membrane, subsequently causing the translucent eggshell phenomenon.
We deeply appreciate your suggestion and recognize its significance in enhancing the quality of our research. In future studies, we plan to develop more effective methods to address the solubility issues of eggshell membranes and ensure accurate quantification of these key proteins, thereby better validating our findings.
Thank you once again for your valuable feedback and attention to our research.
Sincerely
Reviewer 2 Report
Comments and Suggestions for Authors
The authors have done a satisfactory job revising the manuscript.
Author Response
Thank you for your positive feedback on our revised manuscript. We appreciate the time and effort you have invested in reviewing our work and are pleased to hear that the revisions have met your expectations. We look forward to the possibility of our manuscript's publication and are grateful for the guidance provided throughout this process.
Thank you once again for your support and constructive comments.
Best regards